# Potential Effects of Sucralose and Saccharin on Gut Microbiota: A Review

**DOI:** 10.3390/nu14081682

**Published:** 2022-04-18

**Authors:** Susana del Pozo, Sonia Gómez-Martínez, Ligia E. Díaz, Esther Nova, Rafael Urrialde, Ascensión Marcos

**Affiliations:** 1Department of Nutrition and Food Science, Faculty of Pharmacy, Complutense University of Madrid, 28040 Madrid, Spain; 2Department of Metabolism and Nutrition, Institute of Food Science, Technology and Nutrition (ICTAN), Spanish National Research Council (CSIC), 28040 Madrid, Spain; sgomez@ictan.csic.es (S.G.-M.); ldiaz@ictan.csic.es (L.E.D.); enova@ictan.csic.es (E.N.); amarcos@ictan.csic.es (A.M.); 3Department of Genetics, Physiology and Microbiology, School of Biological Sciences, Complutense University of Madrid, 28040 Madrid, Spain; rurriald@ucm.es; 4Area of Nutrition and Bromatology, Department of Pharmaceutical and Health Sciences, San Pablo CEU University, 28925 Madrid, Spain

**Keywords:** saccharin, sucralose, gut microbiota, acceptable daily intake, short-term studies, long-term studies, short-chain fatty acids

## Abstract

Artificial sweeteners are additives widely used in our diet. Although there is no consensus, current evidence indicates that sucralose and saccharin could influence the gut microbiota. The aim of this study was to analyze the existing scientific evidence on the effects of saccharin and sucralose consumption on gut microbiota in humans. Different databases were used with the following search terms: sweeteners, non-caloric-sweeteners, sucralose, splenda, saccharin, sugartwin, sweet’n low, microbiota, gut microbiota, humans, animal model, mice, rats, and/or in vitro studies. In vitro and animal model studies indicate a dose-dependent relationship between the intake of both sweeteners and gut microbiota affecting both diversity and composition. In humans, long-term study suggests the existence of a positive correlation between sweetener consumption and some bacterial groups; however, most short-term interventions with saccharin and sucralose, in amounts below the ADI, found no significant effect on those groups, but there seems to be a different basal microbiota-dependent response of metabolic markers. Although studies in vitro and in animal models seem to relate saccharin and sucralose consumption to changes in the gut microbiota, more long-term studies are needed in humans considering the basal microbiota of participants and their dietary and lifestyle habits in all population groups. Toxicological and basal gut microbiota effects must be included as relevant factors to evaluate food safety and nutritional consequences of non-calorie sweeteners. In humans, doses, duration of interventions, and number of subjects included in the studies are key factors to interpret the results.

## 1. Introduction

Humans are drawn to sweetness, but the WHO directives state that free sugars should not represent more than 10% of the daily caloric contribution and propose a reduction to 5% [1]. Sweeteners are substances used to impart a sweet taste to foods either in food manufacturing or as tabletop sweeteners, substituting for sugars. Nowadays, they are much more abundant than they used to be in some types of popular foods consumed by adults and children, because of their lower calorie content [2,3]. They are used in very small amounts and either do not provide any calories or provide just a few. Indeed, they replace added sugars in a wide variety of foodstuffs [4]. For example, in the Spanish market the distribution of food and beverage subgroups (%) containing one or more low- and no-calorie sweeteners comprises bakery and pastry (16%); yogurt and fermented milks (10%); chewing gums, candies, and sweets (10%); food supplements and substitutes (9%); diet soft drinks (7%); sugar soft drinks (7%); sausages and other meat products (6%); and others [5].

Intensive sweeteners have a negligible caloric contribution and high sweetening capacity, higher than sucrose, thus only being necessary in very low doses to obtain intense sweetness because of their high affinity for the tongue papillas. Sweeteners, like all other food additives, are subjected to strict safety control. There are currently 19 compounds authorized for use in food products by the European regulations, 7 of them being classified as polyols (low-calorie sweeteners) and the remaining 12 as non-calorie sweeteners, of which the most notable ones are acesulfame K (E950), aspartame (E951), cyclamates (E952), saccharin (E954), sucralose (E955), neotame (E961), and steviol glycosides (E960) [6]. These compounds have very different chemical structures, although all of them have in common the ability to potently activate some of the multiple potential ligand-binding sites of the sweet-taste receptors in human subjects [7]. In fact, with health concerns regarding currently available sweeteners, there is renewed interest in identifying a safe and palatable sweetener [8]. In addition, sweeteners, like any other element in the diet, can influence the gut microbiota [9].

The human body is inhabited by trillions of symbiotic microorganisms, most of which are found within the gastrointestinal tract, mainly in the large intestine, and they are collectively called the microbiota [10,11]. The gut microbiota are composed of several species of microorganisms, including more importantly bacteria, archaea, yeasts, and viruses, each individual being provided with a unique gut microbiota profile [12]. Eubiosis, the term used for a “healthy microbiota” can be considered the balance of the intestinal microbial ecosystem, with a preponderance of potentially beneficial bacteria species [13]. In opposition, an altered balance is termed dysbiosis. The optimal healthy gut microbiota composition is different for each individual [12]. Human gut microbiota depend on several factors, such as the type of birth (vaginal/caesarean), breast-feeding or bottle-feeding, type of dietary intake, especially during the first two years of life, as well as the environmental living conditions. This is called the basal commensal microbiota. However, microbiota continue to evolve and adapt throughout the whole life of each individual, taking into account certain factors, such as diet, eating behavior, physical activity, sedentary habits, weight and stress management, as well as sleep quality and quantity [14]. The Microbiome Project revealed that there are 600,000 microbial genes in the human gastrointestinal tract. Ninety-nine percent of these are of bacterial origin; the rest are from Archaea and a very small proportion are of viral origin. The core bacterial microbial genes mainly belong to the Firmicutes and Bacteroidetes phyla, followed by Actinobacteria, Proteobacteria, Fusobacteria, and Verrucomicrobia to lesser extents [15]. Typically, restricted anaerobes (such as *Bacteroides*, *Clostridium*, *Eubacterium*, *Ruminococcus*, *Peptococcus*, *Fusobacterium*, and *Bifidobacterium*) prevail over facultative anaerobic genera (such as *Lactobacillus*, *Escherichia*, *Enterobacter*, *Enterococcus*, *Proteus*, and *Klebsiella*), with *Cyanobacteria*, *Fusobacteria*, and *Spirochaeataceae* being less predominant [16].

The composition and activity of the gut microbiota during life is changing and shaped by several factors; most notably, diet and dietary factors are major determinants of gut microbiota composition and activity [14]. The gut microbiota of an individual can reflect his/her diet at any time. A recent study links the state of the gut microbiota and the Mediterranean diet, which was recognized in 2016 as an Intangible Cultural Heritage of Humanity and is associated with the prevention of cardiovascular and metabolic diseases. The study concluded that several beneficial bacteria (*Bifidobacterium animalis*, *Oscillibacter valericigenes*, and *Roseburia faecis*) are more abundant in individuals with greater adherence to the Mediterranean diet [17]. However, the current Western dietary pattern, rich in saturated fats and sugar, is related to an altered composition of the microbiota (often qualifying as less diverse), which seems to be involved in the development of inflammatory metabolic diseases such as obesity or diabetes [18]. Gut microbiota changes correlate with health status [19]. The activity of the gut microbiota in humans includes degradation of undigested proteins and carbohydrates (sugars, oligosaccharides, peptides, amino acids), amino acid and monosaccharide fermentation, hydrogen disposal, bile-acid transformation, and vitamin synthesis [9,20]. Any change in the profile of sugars/sweeteners we consume redefines the nutrient environments in our gut. How indigenous and exogenous microbes use these environments can result in benign, detrimental, or beneficial effects on the host [16].

Until a few years ago, non-caloric sweeteners were considered metabolically inert and without apparent physiological effects; however, some of them undergo multiple changes in the intestine, interacting with the gut microbiota and thus modifying their metabolites in different regions of the intestine [17]. Some studies have reported that sweeteners may have the ability to modify the gut microbiota [7,11,18,19,20,21]. Some of the previously published review works on sweeteners and gut microbiota indicate that, considering experimental studies and clinical trials in human, among the non-nutritive sweeteners, only saccharin and sucralose change gut microbiota populations [2,10,22], so in this review we will focus on these two sweeteners.

Saccharin (E 954) brand names include Sweet and Low^®^, Sweet Twin^®^, Sweet’N Low^®^, and Necta Sweet^®^ [23]. In 1878, saccharin was the first intense sweetener discovered, being potassium, sodium, and calcium salts the most used. Taking sucrose as a reference, its sweetening power is 300–500 [24] and it does not provide any calories. A range of foods and beverages are sweetened by saccharin [2].

The acceptable daily intake (ADI) for saccharin and its sodium, potassium, and calcium salts, that is, the amount of food additive expressed on a body weight basis, established by the Joint FAO/WHO Expert Committee on Food Additives (JECFA) and the Scientific Committee on Food (SCF), is 5 milligrams per kilogram of body weight per day (mg/kg/d) [25] while other agencies are more restrictive, such as ANMAT, which indicates 2.5 mg/kg/d [26]. This is the amount that can be consumed daily throughout life without appreciable health risks (Table 1) [27].

The study of its effect on the gut microbiota began at the end of the last century [11,23,32,33]. Saccharin is mostly absorbed in the stomach, with approximately 85% to 95% of ingested saccharin absorbed and eliminated in the urine, and the remainder excreted in the feces [22,24]. Only 15% of the consumed saccharin makes contact with the colonic microbiota, which suggests that only when consumed in high doses could it alter the intestinal microbiota composition [22].

Sucralose (E 955), FSA-Q-2011-00724, was discovered in 1976. Sucralose is sold under the brand name Splenda^®^ [23]. Sucralose is a substituted disaccharide, a non-nutritive sweetener that is synthesized by the selective chlorination of sucrose in three of the primary hydroxyl groups [34]. The chemical name for sucralose is 1,6-dichloro-1,6-dideoxy-b-D-fructofuranosyl 4-chloro-4-deoxy-a-D-galactopyranoside [24]. Taking sucrose as a reference, its sweetening power is 600 [24]. Its ADI is 15 mg/kg/d of body weight by the JECFA (Joint Expert Committee on Food Additives) [28], EFSA (European Food Safety Agency) [29], and ANMAT (National Administration of Drugs, Foods and Medical Devices) [26] (Table 1).

Sucralose is poorly absorbed, undergoes little metabolism, and enters unchanged into the lower gastrointestinal tract, being excreted primarily unchanged in the feces in all species, including humans, and more than 85% of the consumed sucralose reaches the colon [23]. Therefore, sucralose could possibly either alter or change the gut microbiota composition, although it is scarcely metabolized by intestinal bacteria [24].

When evaluating the effects of saccharin and sucralose on the gut microbiota, several aspects must be considered, including the dose used in the studies and the average daily amount consumed by the population and the ADI of these sweeteners. In particular, the ADI is used in many studies on gut microbiota and sweeteners as a reference dose. As an example of average consumption by a population, we can take the data on sweetener consumption by the Spanish population. In 2020, 0.11 kg/per capita was consumed, which was 26.2% more than in 2019 [35]. This amount represents 0.3 g/p/d of different sweeteners (Table 2). The ADIs for saccharin and sucralose, according to the JECFA, are 5 mg/kg/day and 15 mg/kg/day, respectively [25,28], which means that a 70 kg subject could consume a maximum of 350 mg of saccharin and 1050 mg sucralose. Based on this, the average consumption of the Spanish population would not exceed the ADI for either of the two sweeteners, but it should be considered that these are average data and there may be people with higher consumptions that are exceeding the ADI. Thus, evaluating how those doses may impact the microbiota composition is not without relevance.

In view of this knowledge on non-caloric sweeteners, the aim of this article was updating the existing evidence on the effect of consuming different amounts of saccharin and sucralose in short- and long-term studies on the composition of the gut microbiota.

## 2. Materials and Methods

A descriptive review was conducted to investigate whether there are potential effects of saccharin and sucralose consumption on gut microbiota composition.

The PubMed, Scopus, Google Scholar, ScienceDirect, and Scielo databases were used for the search. The terms entered in this search were as follows: sweeteners, non-calorie sweeteners, sucralose, splenda, saccharin, sugar-win, sweet’n low, microbiota, gut microbiota, human, animal model, mice, rat, and in vitro studies.

Using the term “sweeteners”, for the last 5 years, 1573 clinical trials, meta-analyses, and randomized controlled trials, together with 2984 reviews and systematic reviews, were found. When narrowing the search also including the term “microbiota”, we found 41 clinical trials, meta-analyses, and randomized controlled trials, plus 144 reviews and systematic reviews.

The following exclusion criteria were used: studies that focused on microbiota other than the gut microbiota, studies that did not include the effect of saccharin and sucralose on the gut microbiota, studies that included supplements and/or prebiotics and/or probiotics that affect the gut microbiota, and studies carried out in populations with diseases.

All these studies were divided into in vitro and in vivo studies, differentiating in the latter between studies in animal models and in humans. Finally, for the present review, 6 in vitro studies were evaluated, plus 14 in vivo studies in animal models and 4 in vivo studies in humans (Figure 1). Of the studies included in this publication, 10 were not present in previous reviews, 2 were studies in humans, 6 were studies in animal models, and 2 were in vitro studies.

The following formula was used to estimate the concentrations of saccharin and sucralose used in the animal studies with respect to the ADI in humans when the work did not indicate this, when it was possible with the published data.

ADI (EFSA/JECFA) (mg/kg/day) × Average animal weight (kg)/Average daily liquid intake (mL) (modified from Suez et al.) [34].

The amount of water consumed by the experimental animals was estimated according to the data indicated by Bachmanov et al. [36,37] and the animal care and use committee of the Johns Hopkins University [37].

### 2.1. Effects of Sweeteners on the Gut Microbiota: In Vitro Trials

In vitro models can be used to study the potential effects of sweeteners, specifically saccharin and sucralose, in humans. Data obtained from in vitro studies can serve as hypothesis generators and as indicators of possible interactions between these sweeteners and the gut microbiota.

In vitro studies focus on the changes in the main microbial groups and selected species together with their metabolites, analyzing the diversity, richness, and abundance in the community over time. The in vitro studies included in this review (Table 3) have commonly addressed the interactions between bacteria, intestinal epithelium, and simulated transit.

In 2018, Harpaz et al., evaluated the relative toxicity for the bacteria of artificial sweeteners, approved by the FDA and in a range of concentrations based on acceptable daily intake (ADI). Genetically modified bacteria (*E. coli*) showing luminescence after exposure to certain stresses were used. Both the induced luminescent signals and bacterial growth were measured. The dose-dependent toxicity effect on *E. coli* in vitro was demonstrated [38]. In addition, Wang et al., (2018) evaluated the bacteriostatic effect of sucralose and saccharin on the growth of *E. coli* in liquid and solid media, finding that the ability to selectively inhibit the growth of enteric bacterial species may be due to inhibition of metabolic enzymes or alterations in nutrient transport [39,44,45].

According to Markus et al., using concentration ranges of non-calorie sweeteners, with comparable concentrations within FDA-approved acceptable daily intake (ADI), aspartame, sucralose, and saccharin are not bactericidal but may affect the bacterial communication system via a molecular system termed quorum sensing (QS)-inhibition and by extension may also affect the host metabolism. According to these authors, this outcome may be due to the significant inhibitory actions of these sweeteners on the Gram-negative bacteria N-acyl homoserine lactone-based (AHL) communication system. However, there is a need to continue to elucidate the mechanisms of action involved in the effects of these sweeteners and other related products on gut microbiota [40].

Gerasimidis et al., in 2020 investigated the effect of artificial sweeteners on the gut microbiome and fiber fermentation capacity. To conduct their study, they fermented fecal samples from 13 healthy volunteers in cultures with sweeteners (aspartame, sucralose, stevia-based sweetener). They measured short-chain fatty acid (SCFA) production by gas chromatography and characterized the composition of the microbiome with 16S rRNA sequencing and quantitative polymerase chain reaction (qPCR). Among their results they found that compared to the control, sucralose (*p* = 0.025) significantly increased valeric acid production and induced significant changes in microbiome community structure (β-diversity); using the Bray–Curtis dissimilarity index, it also increased the relative abundance of *Escherichia/Shigella* species as well as *Bilophila* [41].

However, Shil et al., conducted a study using gut microbiota and epithelial models on the role of commonly consumed sweeteners in the pathogenicity of gut bacteria. The effect of non-calorie sweeteners on *E. coli* and *E. faecalis* growth in planktonic culture was measured in vitro after exposure for 4 days to varying concentrations of non-calorie sweeteners (saccharin, sucralose, and aspartame). All these sweeteners increased the ability of model gut bacteria to adhere to and invade intestinal epithelial cells except for saccharin, which had no significant effect on *E. coli* invasion. Furthermore, a negative effect of these artificial sweeteners has been shown on intestinal epithelial cell apoptosis and permeability, thus further increasing the opportunity for bacteria to traverse the gut epithelium and cause septicemia [42].

Some authors (Vamanu et al., 2019), with the aim of establishing the effect of sweeteners on the microbiota pattern of healthy individuals, used a static in vitro system to simulate the transit through the three segments of the human colon. Under these conditions, both the fermentative response and microbial diversity were found to be altered after treatment with in vitro sweeteners, specifically sucralose and saccharin (equivalent to 9 g of sugar), also showing that non-nutritional sweeteners can induce toxicity, expressed by the establishment of dysbiosis [43].

All the reviewed in vitro studies allow us to hypothesize that in one way or another the consumption of artificial sweeteners can affect the bacteria present in the gut microbiota. We must be careful when interpreting the results and consider different aspects, such as the fact that the in vitro conditions may not correspond to the in vivo conditions of the organism. In addition, the different methodologies used in these studies may make it difficult to interpret the results.

### 2.2. Effects of Sweeteners on the Gut Microbiota in Animal Models

A summary of the “animal” studies analyzed is given in Table 4. Mainly murine species have been studied and the work focuses primarily on the number of total anaerobic and aerobic bacteria, bacterial diversity, the Bacteroidetes/Firmicutes ratio, fecal transplantation, and the effects of maternal intake of sweeteners on offspring in adulthood. In most studies, sweeteners were administered to the animals as part of the drinking water at different concentrations using the ADI for saccharin and sucralose as a reference (Table 4).

One of the first studies on saccharin and the intestinal microbiota was conducted in 1980 by Anderson and Kirkland in rats. They compared the total anaerobic and aerobic microbial populations of the cecum and the proportion of both in male rats fed 0 or 7.5% saccharin sodium, in Purina laboratory chow, for 10 days. After this period, the authors observed that the highest doses of saccharin in cecal content showed an increase in anaerobes and maintenance of aerobes, implying a downward shift in the anaerobic/aerobic ratio [33]. However, Serrano et al., showed that short-term saccharin supplementation with an equivalent dose to the highest acceptable level (JECFA) is insufficient to alter gut microbiota in apparently healthy mice [46].

Conversely, Falcon et al., found that chronic feeding of a commercial non-nutritive sweetened yogurt (0.3% sodium saccharin and sodium cyclamate, Zero-Cal, SP, Brazil) did not induce differences in the bacterial diversity of adult male Wistar rats, compared to animals fed a standard low-fat yogurt supplemented with 20% sucrose [47].

In addition, the study by Abou-Donie et al., (2008) found adverse effects of sucralose on the gut microbiota. Splenda was administered to male Sprague-Dawley rats by oral gavage at 100, 300, 500, or 1000 mg/kg for 12 week, to evaluate the concentration of sucralose administered to these experimental animals. In the current review, an estimation was carried out taking into account the sucralose consumption of an adult rat drinking between 30 and 50 mL of the substance prepared in the study by Abou-Donia et al., according to the concentrations shown above and compared with the ADI (EFSA, JECFA), observing that all the values used exceeded admissible limits for humans. These data show that the consumption of sucralose produces an imbalance in the gut microbiota, specifically in the total numbers of anaerobic and aerobic bacteria that are reduced, with a significant decrease in beneficial anaerobic bacteria such as *Bifidobacteria*, *Lactobacilli*, and *Bacteroides*. In this study, equivalent levels of sucralose (Splenda^®^) in a single drink sweetened with sucralose per day were used [32]. Likewise, another study by Uebanson et al., using different doses of sucralose, found alterations in the microbiota, specifically suggesting that sucralose intake affected in a dose-dependent manner the relative amount of *Clostridium* cluster XIVa [48].

Sánchez-Tapia et al., studied whether the type of sweetener and the presence of a high-fat diet differentially could regulate the gut microbiota. Sucralose was dissolved in water to a concentration of 1.5%. Sucralose increased the Firmicutes abundance showing a decreasing trend in Bacteroidetes, with lower alpha diversity [49]. In this respect, Wang et al., in 2018 performed an 8 week sucralose treatment in mice; they found no changes in alpha diversity, Actinobacteria, and Proteobacteria, but they did find an increase in the abundance of the Firmicutes group [39].

Recently, Zhang et al., in their study with different low doses of sucralose in obese rats, found that ~0.43 mg (0.11 mg/kg translated to human) sucralose increased the relative abundance of Firmicutes but decreased the relative abundance of Bacteroidetes, and that ~0.62 mg sucralose (0.16 mg/kg translated to human) decreased the relative abundance of Firmicutes but increased that of Bacteroidetes. Therefore, the dose of sucralose consumed influenced the Bacteroidetes/Firmicutes ratio. There were no changes in alpha diversity. The authors concluded that the two lower doses of sucralose used in the study might alter the compositions of fecal microbiota [50]. However, in this study, the authors did not use a normal weight control animal model to evaluate the extent to which the establishment of obesity in these rats could modify the results.

Li et al., in 2021, evaluated the bacterial composition at different taxonomic levels in guinea pigs that for 28 days had received saccharin in their drinking water (5 mM). The abundance of Firmicutes tended to decrease in the saccharin-consuming group compared to the control group, while the abundance of Bacteroidetes increased. Therefore, the Bacteroidetes/Firmicutes ratio was affected. In addition, at the family level, the relative abundances of Muribaculaceae and Lactobacillaceae increased in the saccharin group and at the genus level, the relative abundance of *Lactobacillus* increased, while at the family level, the relative abundance of Erysipelotrichaceae and Eubacteriaceae decreased as well as *Ileibacterium* at the genus level [51].

Bian et al., conducted studies in male C57BL/6 J mice with sucralose and saccharin at concentrations equivalent to the ADI for humans (FDA). In 2017, concentrations of sucralose of 0.1 mg/mL [52] and concentrations of saccharin of 0.3 mg/mL administered to male mice [53], in a long-term study for 6 months, were found to induce gut microbiome perturbations, exemplified by the alteration of inflammation-related bacterial pathways and metabolites [52,53].

In 2014, Suez et al., had already demonstrated that the administration of saccharin, sucralose, and aspartame to mice can modulate gut microbiota composition and function, which leads to a higher risk of glucose intolerance, and this is associated with an increase in *Bacteroides* spp. and Clostridiales when performing fecal transplants in germ-free mice from the animals treated with commercial sweeteners. The sweeteners were dissolved in mouse drinking water to obtain a 10% solution: Sucrazit (5% saccharin, 95% glucose), Sucralite (5% Sucralose), Sweet’n Low Gold (4% Aspartame). As controls, 10% glucose and 10% sucrose solutions were used [34].

In relation to the possible effect of sweeteners on the offspring, Dai et al., in 2020 investigated the effects of maternal sucralose (MS) intake on the offspring susceptibility to suffer from hepatic steatosis in adulthood. C57BL/6 pregnant mice were randomized into an MS group (MS during gestation and lactation) and a maternal control (MC) group (MC diet). MS group mice were given sucralose solution of 0.1 mg/mL, approximately 5–15 mg/kg BW/day, and equal to the upper limit of the FDA ADI. After weaning, all offspring were fed a control diet until 8 weeks of age, and then treated with a high-fat diet (HFD) for 4 weeks. The maternal intake of sucralose was found to inhibit intestinal development, induce intestinal dysbiosis, and decrease the production of butyrate-producing bacteria and butyrate in offspring through downregulation of G-protein-coupled receptor 43 (GPR43), and to exacerbate HFD-induced hepatic steatosis in adulthood. Likewise, at the phylum level, an increase in the relative abundance of Verrucomicrobia and Proteobacteria and a reduction in Bacteroidetes was observed in animals with MS. However, at the genus level, MS increased the abundance of *Akkermansia*, *Blautia*, *Corynebacterium*, and *Robinsoniella*, while, *Alistipes*, *Barnesiella*, *Paraprevotella*, *Saccharibacteria*_genera_inc_ertaesedis, and *Streptococcus* were reduced, with a decrease in alpha diversity [54].

However, we would like to emphasize that after reviewing the studies included in this review, not only the dilution of the sweetener in the drinking water should be considered, but also the adjustment to the amount of water ingested by the animals, because the consumption can vary among different species and strains. For example, the average dose/day of liquid drunk by one mouse can range from 3.9 ± 0.2 mL/mouse to 8.2 ± 0.3 mL [36]. There are also physiological and metabolic differences between rodents and humans [55], and, depending on the type of study and the duration of treatment, inferring the results of investigations using rodent models to those in humans may lead to misleading scientific interpretations. In addition, the metabolism of the sweeteners reviewed in this study can be different between animals and humans, and also among different types of animal species. In fact, in relation to sucralose, there is variability within the types of animals used. However, regarding sucralose (organochlorine), when administered orally, similar results have been found among all species evaluated, showing very low absorption levels and light metabolism. For saccharin, being a water-soluble acid with a pKa of 1.8, absorption is increased in those animal species with lower stomach pH, such as rabbits and humans, compared to those with higher stomach pH, including rats [24]. Thus, studies in animal models are a proxy to studying the potential human effects but human evidence should be gathered at the widest possible extent that the ethics premises in biomedicine and clinical trials may allow.

The animal studies reviewed, except that by Serrano et al. [46], show that saccharin and sucralose produce time- and dose-dependent changes in the gut microbiota. Some studies highlight the modification of the amount of anaerobic and aerobic microbiota, while others emphasize the effect of sucralose on the Bacteroidetes/Firmicutes ratio and others are focused on how maternal consumption can affect the offspring.

However, the mechanisms that mediate the physiological effects of low- or non-calorie sweeteners remain unclear and are most likely diverse. According to the literature, sucralose and saccharin, since they are not absorbed, can influence the maintenance of the pH of the bolus in its trajectory through the intestine, which implies a change in the microenvironmental conditions. Thus, this outcome could be a factor influencing the selective proliferation of certain bacterial groups. In addition, the presence in greater or lesser quantity of cells expressing the T1R2/T1R3 taste heterodimer would be related to the inflammatory effect and possible adaptations of the microbiota [45].

### 2.3. Effects of Sweeteners on the Gut Microbiota in Human Trials

Non-caloric sweeteners (sucralose and saccharin), as food additives, have been evaluated and approved for use in humans by the European Food Safety Authority and subsequently authorized by the European Commission, the Parliament, and the Council of the European Union. Currently, their consumption, as we have already mentioned, is very widespread in the population, especially in hypocaloric foods and diets as an adjuvant for weight loss or in diabetic patients. The fact that their industrial use in a great variety of products has increased favors the non-adverted consumption.

The human studies reviewed, described in Table 5, studied microbial diversity and metabolites, specifically changes in SCFAs, the main metabolites produced by the microbiota in the large intestine [56]. The SCFAs are bacterial metabolites produced during the colonic fermentation of undigested carbohydrates, such as dietary fiber and prebiotics, and can mediate the interaction between the diet, the microbiota, and the host [57]. SCFA levels are influenced by the proportion of intestinal bacteria, whose alteration (dysbiosis) can lead to an unbalanced composition of the gut SCFAs and therefore it has been concluded that supplementation with pure saccharin did not alter microbial diversity or composition [58].

The following are the results of human studies, with a sweetener concentration not exceeding the ADI and short-term intake. Among the intervention studies carried out with saccharin, Serrano et al., performed a double-blind, placebo-controlled, parallel-arm study to explore the effects of pure saccharin compound on gut microbiota and glucose tolerance in healthy men and women (46 subjects completed the study; IMC ≤ 25). Participants were randomized into four treatment groups (placebo, saccharin, lactisole, or saccharin with lactisole) and consumed capsules containing pulp filler/placebo (1000 mg/d) sodium saccharin (400 mg/d), lactisole (670 mg/d), or sodium saccharin (400 mg/d) + lactisole (670 mg/d) twice daily for 2 weeks. The authors concluded that in these conditions, microbial diversity or composition at any taxonomic level were not changed by pure saccharin supplementation in humans. According to these results, short-term saccharin consumption at maximum acceptable levels (JECFA) is not sufficient to alter the gut microbiota or induce glucose intolerance in supposedly healthy humans [46]. However, Suez et al., did find some modifications in the gut microbiota in 4 of 7 healthy volunteers (5 men and 2 women, aged 28–36 years) from an ongoing clinical nutritional study who were selected as non-habitual sweetener consumers. A saccharin intervention was conducted for one week in which they consumed, on days 2 to 7, the FDA maximum acceptable daily intake (ADI) of commercial saccharin, in three daily doses (equivalent to 120 mg). Changes in the microbiota of only 4 participants, who had developed significantly worse glycemic responses in the study, were observed, and they suggest that humans exhibit a personalized response to non-caloric artificial sweeteners, possibly derived from differences in their basal microbiota [34].

In relation to sucralose, Thomson et al., (2019) conducted a randomized, double-blind study in 34 healthy men (18–50 years) with BMI 20–30 kg/m^2^. Sixteen subjects were administered for one week a dose of 780 mg of sucralose per day that was divided into three-260 mg intakes; the control group received a placebo (*n* = 17). In this study, at the phylum level, the gut microbiome was not modified in healthy individuals [60].

Similar results were obtained in a randomized, double-blind, crossover, controlled clinical trial involving the follow-up of 17 healthy participants. They performed a crossover design for 12 weeks (two 14 day treatment periods separated by a 4 week washout period). In weeks 5 and 6, the volunteers consumed aspartame (*n* = 9) or sucralose (*n* = 8). Prior to the washout period, in which no artificial sweeteners were consumed in weeks 11 and 12, all participants consumed the sweetener that they had not previously consumed. The participants were administered 14% (0.425 g) of the ADI for aspartame and 20% of the ADI for sucralose (0.136 g) (approximately 10.5 packets of sucralose with beverages). To define the ADI, they used Health Canada data (sucralose as 9 mg/kg body weight and 40 mg/kg/bw for aspartame). The relative abundance of the five most abundant genus-level taxa within the four most dominant phyla (Actinobacteria, Bacteroidetes, Firmicutes, and Verrucomicrobia) before and after treatment were analyzed at the following days: 1, 28, 42, and 84. Alpha diversity estimation was performed with the Shannon index on the raw operational taxonomic unit. No changes were found for aspartame and sucralose in the gut microbiota composition or SCFAs after 14 days of a daily intake in healthy participants [59].

In relation to long-term studies with saccharin and sucralose in humans, there are not any studies to our knowledge. In the study conducted by Suez et al., in 2014 on the relation between artificial sweetener consumption and gut microbiota, the effect of long-term consumption of non-caloric artificial sweeteners was evaluated. To this end, a validated food frequency questionnaire comprising data collected from 381 non-diabetic individuals from an ongoing clinical nutritional study was used. The results show that artificial sweetener consumption increases the risk of glucose intolerance, these adverse metabolic effects being mediated by modulation of the composition, metabolic function, and the basal microbiota. In this regard, Aldrete-Velasco et al., pointed out in a review that under this design, eliminating completely the confounding variables was not possible, so changes in the microbiota and their metabolic characteristics could also be different due to other factors beyond the consumption of non-caloric sweeteners [61].

Considering the results mentioned above and according to other authors, by using high doses of saccharin and sucralose both in in vitro studies and in animal models, gut microbiota can be modified, whereas in human studies performed using amounts below the ADI and in short-term studies, no effects on gut microbiota are found [2,10,16,47,48,49]. Contrary to this outcome, Schiffman et al., in 2019 stated in an editorial regarding in vivo animal models, involving data on low- and non-caloric sweeteners and gut microbiota, that sucralose can unequivocally and irrefutably alter the gut microbiome at those levels approved by regulatory agencies, associated with human use. These authors also highlight that it is not appropriate to draw generalized conclusions about effects on the gut microbiota [62].

According to several studies, the explanation for these results may be due to the different doses used in in vitro and in animal model studies versus in human studies, where the doses are lower than the ADI [16,48]. In addition, in human clinical studies, the sample sizes are small, as well as the duration of the interventions. In addition, there is a relevant point to bear in mind like the failure in considering the knowledge regarding the basal gut microbiota of volunteers.

## 3. Conclusions

In conclusion, it is necessary to broaden the concept of food safety for sucralose and saccharin by re-evaluating toxicity referring to the effect on the gut microbiota and the possible consequences on health maintenance and disease amelioration in humans. Indeed, the mechanisms by which low-calorie and non-calorie sweeteners may alter the gut microbiota remain unclear, and it is not possible to conclude at present whether their effect is direct on the microbiota or mediated by the metabolic situation of the host, for which there are still no conclusive studies. In fact, the scientific literature in both health and disease sometimes refers to beneficial strains and other studies focus on pathogenic strains, which may be due to the lack of clarity regarding what defines dysbiosis or eubiosis. In order to obtain sufficient evidence in these types of studies, clinical trials should be conducted bearing in mind an adequate number of subjects, as well as considering their baseline gut microbiota, dietary habits, and lifestyles. Although the preferred population is healthy adults due to its easy accessibility, more studies must be conducted taking vulnerable population groups into account, such as children, the elderly, pregnant women, lactating women, or subjects with intestinal pathologies, obesity, diabetes, cardiovascular diseases, etc. and chronic and/or excessive consumers of low- and non-calorie sweeteners.

## Figures and Tables

**Figure 1 nutrients-14-01682-f001:**
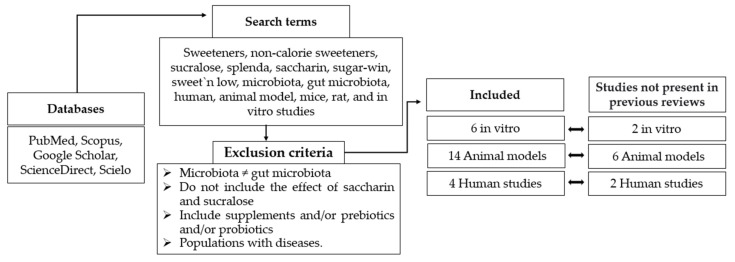
Flow chart regarding selection method.

**Table 1 nutrients-14-01682-t001:** Acceptable daily intake (ADI) (mg/kg/bw).

	JECFA ADI [25,28]	EFSA ADI [29,30]	Health Canada	[31] ANMAT [26]
Saccharin	5	5	5	2.5
Sucralose	15	15	9	15

**Table 2 nutrients-14-01682-t002:** ADI. Mean consumption of sweeteners in the Spanish population.

	Saccharin	Sucralose
ADI mg/kg body wt (JECFA)	5 mg/kg	15 mg/kg
ADI subject 70 kg	350 mg	1050 mg
Average consumption of the Spanish population	300 mg/day	

**Table 3 nutrients-14-01682-t003:** Summary of the analyzed in vitro studies.

Reference	Sweeteners/Doses/Duration	Methods	Bacteria	Results/Conclusions Saccharine and/or Sucralose
Harpaz et al., 2018 [38]	Aspartame, sucralose, saccharine, neotame, advantame, and acesulfame potassium-k (ace-k). ADI (FDA)	Bioluminescent	*E. coli* strains (TV1061, DPD2544 and DPD2794)	Toxic effects
Wang et al., 2018 [39]	Sucralose, saccharin, acesulfame potassium, and rebaudioside Liquid assay: equal molarity of sodium chloride/5 h Agar: 1.25% (*w*/*v*) sucralose and 2.5% (*w*/*v*) sucralose/24 h	Liquid culture assay. LB agar plate assay	*E. coli* HB101 and *E. coli* K-12	Bacteriostatic effects
Markus V, et al., 2021 [40]	Aspartame, sucralose, saccharin Bioluminescence assay, growth assay: 10 µL non-calorie sweeteners or sports supplements. Swarming motility assay: aspartame (1.36 mM), sucralose (25.2 mM), or saccharine (2.72 mM) QS competition assay using Chromobacterium Violaceum CV026/20 h	Biosensor assays, biophysical protein characterization methods, microscale thermophoresis, swarming motility assays, growth assays, and molecular docking	*E coli* K802NR and *P. aeruginosa* lasRI *P. aeruginosa* PAO1 *C. violaceum* (CV026)	Inhibition of quorum sensing
Gerasimidis C et al., 2020 [41]	Aspartame-based sweetener, sucralose, stevia 50% ADI (male, w: 75 kg)	Gas chromatography	Total bacteria (feces from healthy individuals) and 5 bacterial groups (Bacteroides/Prevotella, *Bifidobacterium, B. coccoides, C. leptum* and *E. coli*)	Sucralose: shifted microbiome community structure ↔ bacterial populations ↑ Escherichia/Shigella
Shil A and Chichger, H, 2021 [42]	Saccharin, sucralose, and aspartameGrowth curve: 0.1 to 1000 µM/4 d Biofilm formation assay: 100 µM/48 h Haemolysis assay, adhesion assay, and invasion assay: 100 M/24 h Cytotoxicity assay: 100 M/48 h	Models of microbiota and the intestinal epithelium (Caco-2 cells)	*E. coli* NCTC10418 and *E. faecalis* ATCC19433 *S. aureus*	Saccharin bacteriostatic effects Saccharin, sucralose: ↑ biofilm formation ↑ ability of bacteria to adhere to, invade, and kill gut epithelial cells (exception saccharin on *E. coli*) Negative effect on intestinal epithelial cell apoptosis and permeability
Vamanu E et al., 2019 [43]	Sodium cyclamate, sucralose, sodium saccharin, steviol, white sugar 40 mg active substance (more than 90% purity)	Static GIS1 simulator (three segments of the human colon)	Total microbial (feces from healthy individuals)	Saccharin: ↓ number of microorganisms; ↓ SCFAs Both: ↓ phylum Firmicutes; ↓ fermentative processes; ↑ colonic pH; ↑ 10% ammonia synthesized; ↓ SCFAs

ADI: acceptable daily intake; SCFA: short-chain fatty acid. ↔: unmodified; ↑: increase; ↓: decrease.

**Table 4 nutrients-14-01682-t004:** Evidence from animal model studies relative to sucralose and saccharin effects on the gut microbiota.

Reference	Sweeteners/Doses/Duration	Animal Model	Results
Anderson & Kirkland, 1980 [33]	Treatment: 7.5% sodium saccharin in the Purina laboratory chow Control: Cellulose 7.5% in the Purina laboratory chow Duration: 10 d	Weaning male Charles River rats (Weight 55 ± 3 g) (*n* = 7)	↑ The numbers of aerobic microbes ↓ Anaerobic/aerobic ratio
Serrano et al., 2021 [46]	Treatment: saccharin average daily dose equal to 4 times (250 mg/kg) the human ADI (JECFA) Control: water Duration: 10 wk	8-wk-old mice	↔Alpha and beta diversity and relative microbial abundances
Falcon et al., 2020 [47]	Control: Sucrose-sweetened yogurt (suc): low-fat yogurt supplemented with 20% sucrose, final solution concentration 11.4% sucrose Treatment: NNS-supplemented yogurt: (0.3% sodium saccharin and sodium cyclamate). Final solution concentration 0.17% NNS Duration: 17 wk	Adult male Wistar rats (weight: 210 ± 6 g) SUC (*n* = 9 per group) NNS (*n* = 10 per group)	↔Species richness ↔ Shannon or Simpson diversity indices
Abou-Donia et al., 2008 [32]	Treatment: Splenda (Sucralose) oral gavage: 1.1; 3.3; 5.5 and 11 mg/kg/d sucralose concentrations. Control: water Duration: 12 wk	Male Sprague-Dawley rats (weight: 200–240 g) (*n* = 10 per group)	*↓* Number of total anaerobes and other anaerobic bacteria (*Bifidobacteria, Lactobacilli, Bacteroides, and Clostridium*).
Uebanso et al., 2017 [48]	Treatment: LS (sucralose solution of 1.5 mg/kg bw/d). HS (sucralose solution of 15 mg/kg bw/d), which is equal to the maximum ADI. Control: distilled water Duration: 8 wk	Male and female C57Bl/6 J mice (4 wk old) (*n* = 8)	LS vs. HS ↔The relative amounts of fecal total bacteria LS vs. HS ↔ Firmicutes and Bacteroidetes phylum bacteria ↓ relative *Clostridium* cluster XIVa, dose-dependent
Sánchez-Tapia et al., 2020 [49]	Treatment: Sucralose: drinking water 1.5% sucralose Control: water Duration: 4 mo	Male Wistar rats (5 wk old) (*n* = 6 per group)	↓α-diversity ↑ *B. fragilis* abundance
Wang et al., 2018 [39]	Treatment: Sucralose: drinking water sucralose (2.5%, *w*/*v*) Duration: 8 wk	C57BL/6 mice (5 wk old)	↔ α-diversity, Actinobacteria, and Proteobacteria ↑ Abundance of Firmicutes
Zhang et al., 2021 [50]	Treatment: daily gavage of Sucralose ∼ 0.43 mg, sucralose ~0.62 mg. Control: daily gavage of 2 mL normal saline Duration: 4 wk	Obese Sprague Dawley rats (4 wk old) (8 weeks after high fat diet (HFD)) (*n* = 6 per group)	0.43 mg sucralose: ↑ relative abundance of Firmicutes and ↓ Bacteroidetes 0.62 mg sucralose: ↓ relative abundance of Firmicutes ↑ Bacteroidetes The ratio of Firmicutes to Bacteroidetes in 0.43 mg sucralose was higher than that in 0.62 mg
Li et al., 2021 [51]	Treatment: Saccharin sodium in drinking water: 1.5 mM Control: water Duration: 4 wk	Female Harley-white guinea pigs (Cavia porcellus) (4 wk old) (weight: 240.7 ± 7.7 g) (*n*= 6 per group)	↑ Firmicutes and Lactobacillasceae-Lactobacillus abundance ↓ Relative abundance of *Erysipelotrichaceae*, *Eubacteriaceae*, and *Ileibacterium*
Bian et al., 2017 [52]	Treatment: Sucralose tap water (0.1 mg/mL). ADI (FDA) Control: tap water Duration: 6 mo	C57BL/6 male mice (~8 wk old) (*n* = 10 per group)	↑Numerous bacterial toxin genes (toxic shock syndrome toxin-1 and shiga toxin subunits) 14 genera exhibited different patterns over time in sucralose, different after 3 and/or 6 mo of treatment
Bian et al., 2017 [53]	Treatment: Saccharin, drinking water (0.3 mg/mL). ≈ ADI (FDA) Control: tap water Duration: 6 mo	C57BL/6 J male mice (Weight, ~23 g, ~8 wk old) (*n* = 10 per group)	Alterations of the gut metabolome with 1743 significant changes in molecular features 3 mo: ↑*Sporosarcina, Jeotgalicoccus*, *Akkermansia*, *Oscillospira*, and *Corynebacterium* *↓Anaerostipes* and *Ruminococcus* 6 mo: *↑Corynebacterium*, *Roseburia*, and *Turicibacter* *↓Ruminococcus*, *Adlercreutzia*, and *Dorea*
Suez et al., 2014 [34]	Treatment: Commercial NAS in drinking water 10% solution: (5% saccharin, 95% glucose), (5% Sucralose), (4% Aspartame). Pure saccharin (0.1 mg ml^−1^) in drinking Control: water or water with 10% glucose or 10% sucrose Duration: 11 wk NAS and 5 wk pure saccharin	Lean C57Bl/6 mice (10 wk old) with NAS treatment (*n* = 5 per group) C57Bl/6 mice fed on HFD with saccharin treatment (10 wk old) (*n* = 8 per group)	Saccharin: dysbiosis reflected by more than 40 operational taxonomic units (OTUs) abundances changed ↑ *Bacteroides* genus and Clostridiales order Dysbiosis in mice that consumed pure saccharin and HFD
Dai et al., 2020 [54]	MS treatment: gestation and lactation, sucralose 0.1 mg/mL (FDA ADI) Offspring treatment: weaned pups fed a control diet until 8 wk of age and treated with HDF for 4 wk Control: distilled water in MS maternal control and offspring fed with a control diet Duration: maternal treatment, 6 wk	C57BL/6 pregnant mice 3 wk old, weaned pups	MS: at phylum level ↑ the relative abundance of Verrucomicrobia and Proteobacteria and ↓Bacteroidetes At genus level ↑ abundance of *Akkermansia*, *Blautia, Corynebacterium*, *Robinsoniella*, and ↓ *Alistipes, Barnesiella*, *Paraprevotella*, *Saccharibacteria* genera incertae sedis, and *Streptococcus* MS alters the gut microbiota in the offspring, ↓alpha diversity of 3-wk-old pups

ADI: Acceptable daily intake; MS: maternal sucralose; d: day; wk: weeks, mo: months; HFD: high-fat diet; LS: low saccharin; HS: high saccharin. ↔: unmodified; ↑: increase; ↓: decrease.

**Table 5 nutrients-14-01682-t005:** Summary of the analyzed in vivo studies. Humans.

Reference	Sweeteners/Doses/Duration	Design	Results/Conclusions Saccharin and Sucralose
Serrano et al., 2021 [46]	Saccharin 400 mg/d/2 wk	Randomized, double-blind, placebo-controlled interventional study	↔gut microbiota
Ahmad et al., 2020 [59]	Sucralose and aspartame 20% ADI sucralose (~0.136 g sucralose)/14 d	Randomized, double-blind crossover (12 wk) and controlled clinical trial.	↔ gut microbiota ↔ SCFAs
Thomson et al., 2019 [60]	Sucralose 780 mg/d/7 d	Randomized, double-blind study	↔ gut microbiota
Suez et al., 2014 [34]	Saccharin FDA maximal ADI/7 d	Intervention study	Response according to basal microbiota

ADI: Acceptable daily intake; SCFA: short-chain fatty acid; d: day; wk: weeks. ↔: unmodified

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
