# Peer review of "Potential Effects of Sucralose and Saccharin on Gut Microbiota: A Review"

_nutrients, 2022, doi:10.3390/nu14081682_

Round 1

Reviewer 1 Report

This is a timing review of the effects of artificial sweeteners and brings interest to the research community and the public.  However, this review is more descriptive and lacks some conclusions and perspectives for current and further studies.  For example, what are the potential mechanisms of action for the sweeteners modulating gut microbiota?  How do explain the differences in their effects on gut microbiota?  Also, the writing should be much improved for readability. 

Author Response

Dear reviewer, thank you for reviewing and considering for publication our manuscript entitled “Potential effects of sucralose and saccharin on gut mi-crobiota. A review”, for publication in the Nutrients.

Reviewer 2 Report

The authors aimed to review the existing scientific evidence regarding the effect of saccharin and sucralose consumption on gut microbiota. My comments are shown below.

  1. I’m wondering how those artificial sweeteners manipulate gut microbiota (directly or in directly?). Alternation of bacterial quorum sensing might be one of the potential mechanisms but is there any other mechanisms reported? 
  2. Can altered bacterial composition caused by them influence our health? Please discuss it in more detail.
  3. As this review selected only 24 studies for analysis, selection method would be important. I think that a flow chart regarding selection methods would be very helpful and more suitable than the Table 3.
  4. I also request creating a figure summarizing the contents of this review

Author Response

Dear reviewer, thank you for reviewing and considering for publication our manuscript entitled “Potential effects of sucralose and saccharin on gut mi-crobiota. A review”, for publication in the Nutrients
